# Hypertensive Disorders of Pregnancy (HDP) and the Risk of Common Cancers in Women: Evidence from the European Prospective Investigation into Cancer (EPIC)-Norfolk Prospective Population-Based Study

**DOI:** 10.3390/cancers12113100

**Published:** 2020-10-23

**Authors:** Zahra Pasdar, David T. Gamble, Phyo K. Myint, Robert N. Luben, Nicholas J. Wareham, Kay-Tee Khaw, Sohinee Bhattacharya

**Affiliations:** 1Ageing Clinical & Experimental Research (ACER) Team, Institute of Applied Health Sciences, University of Aberdeen, Aberdeen AB25 2ZD, UK; z.pasdar.17@abdn.ac.uk (Z.P.); david.gamble@abdn.ac.uk (D.T.G.); 2Department of Public Health and Primary Care, Clinical Gerontology Unit, University of Cambridge, Cambridge CB2 2QQ, UK; robert.luben@phpc.cam.ac.uk (R.N.L.); kk101@medschl.cam.ac.uk (K.-T.K.); 3MRC Epidemiology Unit, Institute of Metabolic Science, University of Cambridge School of Clinical Medicine, Cambridge CB2 0QQ, UK; Nick.Wareham@mrc-epid.cam.ac.uk; 4Aberdeen Centre for Women’s Health Research, Institute of Applied Health Sciences, University of Aberdeen, Aberdeen AB25 2ZL, UK

**Keywords:** hypertension during pregnancy, pregnancy complications, cancer

## Abstract

**Simple Summary:**

One in ten pregnancies is complicated with high blood pressure. Whilst the association between high blood pressure during pregnancy and cardiovascular disease in later life is well-established, its association with cancer is less so. We aimed to determine the association between hypertensive disorders of pregnancy (HDP) and common cancers in a UK population-based longitudinal study. No statistically significant association was found between HDP and the odds of future cancer, and independent site-specific cancers of breast, colorectal, lung, ovarian and endometrial cancers in a UK-based general practice-derived population. Current research underlining HDP and cancer risk is limited to specific populations only. Future work in different populations is required, focusing on complex genetic and hormonal factors involved in these conditions.

**Abstract:**

Purpose: The purpose was to determine the association between HDP and cancer in a UK cohort. Methods: Between 1993 and 1997, participants from the EPIC-Norfolk cohort attended baseline health-checks and completed questionnaires, where a history of HDP was collected. Incident cancer cases were identified through NHS record linkage until March 2016. Univariable and multivariable logistic regression analyses were employed to determine the association between HDP and odds of cancer, with adjustment for potential confounders including co-morbidities, sociodemographic, lifestyle and reproductive factors. Results: 13,562 women were included after excluding prevalent cancer cases and women with no pregnancies. 2919 (21.5%) reported HDP and 2615 incident cancers occurred during mean follow up of 19 years. Median age (IQR) at baseline for incident cancer was 60.8 (±14.8) years. Among incident cancer cases, 578 (22.1%) had HDP. In multivariable analyses, HDP had odds ratio (OR) 1.06; 95% CI 0.95–1.18 for incident cancer. The ORs (95% CIs) for common site-specific cancers including breast, colorectal, lung, ovarian and endometrial cancers were 1.06 (0.88–1.28), 1.15 (0.92–1.45), 0.96 (0.68–1.35), 1.30 (0.93–1.83) and 1.16 (0.80–1.67). Conclusion: We found no association between HDP and cancer risk. Further studies are required to confirm and account for any underlying genetic factors involved in pregnancy-related exposures and cancer risk.

## 1. Introduction

Hypertensive disorders of pregnancy (HDP) are a major cause of maternal morbidity and mortality globally [1,2], complicating up to 10% of pregnancies worldwide [3]. HDP include four clinical subgroups: (1) preeclampsia–eclampsia, (2) gestational hypertension, (3) chronic hypertension and (4) chronic hypertension with superimposed preeclampsia [3]. HDP are known to influence maternal health in the long-term and it is well-established by a number of systematic reviews and meta-analyses that these women suffer from increased risk of future cardiovascular disease [4,5].

However, less is known regarding HDP and cancer risk. Some studies have shown that HDP, mainly preeclampsia, reduces future breast cancer risk. One recent population-based case-control study of 116,196 breast cancer cases illustrated a decrease in odds of breast cancer, in relation to a history of preeclampsia and gestational hypertension [6]. Other various studies have similarly shown reduced odds or risk of breast cancer, with approximately 10–20% risk reduction [7,8,9,10,11,12,13]. Despite this, research has produced conflicting results. Three meta-analyses have shown no association or inconclusive results [4,14,15] with similar findings reported in large primary studies [16,17,18,19]. In contrast, two cohorts and a case-control study showed increased cancer risk [20,21,22].

Most research has focused on preeclampsia alone [7,9,11,12,17,19,20,21] and have restricted findings to breast cancer. Therefore, the effect of any HDP on other common cancers remains less clear. As an aging population, cancer is a major cause of mortality in women globally [23] with breast, colorectal and lung being the most common cancers in females [24]. Further understanding in this field is vital to reduce the global impact of cancer in women and guide future preventative strategies.

We therefore aim to determine the association between HDP and future cancer risk and its independent association in common female cancers: breast, colorectal, lung, ovarian and endometrial, using prospective population-based data from the European Prospective Investigation into Cancer (EPIC)-Norfolk cohort.

## 2. Results

Among 16,740 women, 13,562 women had no prevalent cancer at baseline and had a pregnancy history. Of these, 2615 (19.3%) had incident cancer diagnosis. Table 1 details comparison of sample characteristics between those with and without incident cancer. Compared with participants who did not have cancer, participants who did were older (60.8 ± 14.8 vs. 57.3 ± 16.2, *p*-value < 0.001), more likely to have no educational qualification (48.7% vs. 44.1%, *p*-value < 0.001), be physically inactive (67.5% vs. 61.6%, *p*-value < 0.001) and belong to the pre-obesity category (35.8% vs. 32.2%, *p*-value < 0.001).

Table 2 details unadjusted and adjusted odds ratios of risk factors for incident cancer with corresponding 95% confidence intervals (95% CIs). HDP had no association with cancer in both univariable and multivariable analyses. In unadjusted analyses, HDP had OR 1.05; 95% CI 0.95–1.17. In the adjusted analysis, this did not change appreciably; 1.06; 95% CI 0.95–1.18.

Among reproductive variables, diabetes during pregnancy had no association with cancer risk (adjusted OR 1.27; 95% CI 0.91–1.78). Furthermore, increasing age at first live birth increased odds of cancer. An age of ≥31 years was associated with the greatest odds (adjusted OR 1.26; 95% CI 1.04–1.52). Having children was not significantly associated with cancer risk in both univariable and multivariable analyses. Additionally, any stillbirths (adjusted OR 0.94; 95% CI 0.73–1.20) and miscarriages and abortions (adjusted OR 0.98; 95% CI 0.70–1.37 for ≥3 miscarriage and abortions) showed no significant association.

As expected, current smokers had greater odds of cancer (adjusted OR 1.57; 95% CI 1.37–1.79) compared to ex-smokers (adjusted OR 1.08; 95% CI 0.98–1.19). Being physically inactive was also significantly associated with the odds of cancer (OR 1.30; 95% CI 1.19–1.42). After adjustment, this effect was slightly attenuated (adjusted OR 1.12; 95% CI 1.02–1.23). Among different body mass index (BMI) categories, belonging to obesity class II or more conferred the greatest odds of cancer (adjusted OR 1.73; 95% CI 1.38–2.18). Prevalent stroke significantly decreased the odds of cancer (adjusted OR 0.54; 95% CI 0.33–0.87).

Table 3 details site-specific unadjusted and adjusted odds ratios for HDP and incident breast, colorectal, lung, ovarian and endometrial cancers, respectively. The descriptive statistics and odds ratios of risk factors for these have been demonstrated as part of the online Appendix A (Appendix A). In this site-specific cancer analyses, HDP was not associated with any of these common female cancers. In the breast cancer analysis, the unadjusted OR was 1.09; 95% CI 0.91–1.31 and the adjusted OR was 1.06; 95% CI 0.88–1.28. For colorectal cancer, univariable analysis revealed an OR of 1.17; 95% CI 0.93–1.45. In the multivariable analysis, this did not vary much (adjusted OR 1.15; 95% CI 0.92–1.45). The lung cancer analysis gave an unadjusted OR of 0.84; 95% CI 0.60–1.18 and an adjusted OR of 0.96; 95% CI 0.68–1.35. Finally, the unadjusted ORs for ovarian and endometrial cancers were 1.36; 95% CI 0.98–1.90 and 1.42; 95% CI 0.99–2.03. In the multivariable analyses, these results were not statistically significant: 1.30; 95% CI 0.93–1.83 and 1.16; 95% CI 0.80–1.67, respectively.

The sub-analyses confining to those with high BMI did not show any overall significant associations (Appendix A, Appendix A).

## 3. Discussion

In this prospective population-based study with 19 years of mean follow-up, we found that HDP was not associated with overall maternal cancer risk. Similarly, the site-specific cancer analyses revealed that HDP was not associated with individual risks of common female cancers including breast, colorectal, lung, ovarian or endometrial cancers. Additionally, our results reinforce that lifestyle factors in the older age are important contributors to cancer risk.

These results are supported by findings from previous studies. A systematic review and meta-analysis on preeclampsia and risk of cardiovascular disease and cancer in later life yielded no association between preeclampsia and breast cancer risk with relative risk (RR) 1.04 (95% CI 0.78–1.39), although significant heterogeneity was observed. Similarly, analysis on any cancer outcomes revealed no association; RR 0.96 (95% CI 0.73–1.27); (*p* = 0.15, I^2^ = 43.2%) [4]. Moreover, two recent meta-analyses reported inconclusive results between preeclampsia and breast cancer risk. A pooled hazard ratio (HR) of 0.86 (95% CI 0.73–1.01) was reported in one study [14] and a risk ratio (RR) 0.93 (95% CI 0.82–1.06) was observed in another [15]. No association was also reported for pregnancy-induced hypertension in both meta-analyses; HR 0.83 (95% CI 0.66–1.06) [14] and RR 0.95 (95% CI 0.81–1.12) [15]. Our findings have been corroborated by several primary studies as well [16,17,19]. A Swedish cohort of 314,019 women found no association between pregnancy complications including hypertensive disorders of pregnancy and breast cancer risk [16]. Another cohort of 40,951 women revealed no association between preeclampsia and cervical, endometrial, ovarian and breast cancer risk [19]. However, a recent Nordic population-based case-control study of 116,196 breast cancer cases showed decreased breast cancer risk in women with a history of preeclampsia (OR 0.91; 95% CI 0.88–0.95) or gestational hypertension (OR 0.90; 95% CI 0.86–0.93) after adjusting for maternal birth year, country and parity [6]. Similar results of cancer risk reduction have been echoed in other studies based on American, Swedish, Scottish and Norwegian populations [7,8,9,10,11,12,13]. There have also been reports of increased cancer risk in three studies [20,21,22]. Two Israeli cohorts showed an age-adjusted HR of 1.27 (95% CI 1.03–1.57) [20] and 1.23 (95% CI 1.05–1.45) [21] for any cancer outcomes according to a history of preeclampsia. In addition, an Italian case-control study revealed an OR of 1.8 (95% CI 1.0–3.4) for breast cancer in relation to a history of hypertension in pregnancy [22]. These differences in findings may possibly be explained by underlying differences in genetic factors in different populations. Therefore, it is possible that underlying genetic variability between study populations may lead to variation in results. Further studies are required to be conducted amongst different populations and additional gene analysis may test this hypothesis. Like cancer, preeclampsia proves to have a genetic component. It is thought that multiple gene loci can contribute towards an individual’s susceptibility with additional environmental factors influencing the expression of these genes, such as maternal age and weight [25]. Differences in sample sizes, environmental factors or differential risks between populations can also contribute to dissimilarities in results. An updated systematic review and meta-analysis would be required to evaluate conflicting results on HDP and cancer risk.

There are some theories explaining risk reduction in individuals with HDP—in particular, preeclampsia—and future breast cancer risk. Though the precise sequence of events is still unclear, the current hypothesis states that lower circulating oestrogen levels occur in preeclampsia. Lower oestrogen levels are thought to accompany higher androgen levels in blood due to insufficient conversion of androgens into oestrogen by a dysfunctional placenta [14]. Oestrogen is considered to be a major risk factor for breast cancer. Preeclampsia presents with an anti-angiogenic profile which may indicate a protective effect against cancer, as tumour growth and metastases require angiogenesis [6,26]. Despite this, there has been conflicting evidence on the lower oestrogen levels in preeclamptic women [27,28].

Our results show that individuals with high BMI (pre-obesity category (adjusted OR 1.26; 95% CI 1.13–1.40) or obesity class II or above (adjusted OR 1.73; 95% CI 1.38–2.18)) are at significantly increased risk of cancer. Although it would be expected to observe a dose–response relationship with odds of cancer increasing with each subsequent rise in BMI category, this was not the case. The adjusted OR for the obesity class I category for incident cancer was 1.10 and this result was not statistically significant (95% CI 0.94–1.27). Belonging to the pre-obesity category significantly increased cancer risk for all cancers in the site-specific analyses, except for lung and ovarian cancers. The highest risk observed was between the obesity class II and above category and endometrial cancer (adjusted OR 5.90; 95% CI 3.36–10.37). Similar findings have been reported previously [29]. In this cohort, the majority of women with HDP belonged to the pre-obesity category (1003 women, 34.4%). Although we showed no statistically significant association between HDP and odds of cancer (adjusted OR 1.06; 95% CI 0.95–1.18), the slight increased likelihood of cancer portrayed by this result may be due to the fact that HDP and cancer share high BMI as a common risk factor [30,31]. It may be that individual lifestyle factors adopted by women with hypertensive pregnancies can override protective hormonal effects that such disorders may have.

Our study has several strengths. We used a large population-based cohort with adequate follow-up time. Detailed data collection by trained personnel adhering to EPIC protocols increases the reliability of our results. Bias is likely to be low in a prospective study with validated outcomes of cancer. Furthermore, we have adjusted for many of the potential risk factors for cancer. Our results are consistent with the literature in reinforcing many well-established risk factors for cancer.

There are some limitations worth acknowledging. Our study relied on self-reported questionnaires for identification of HDP. Maternal recall of this obstetric history has been reported to have high specificity, although sensitivity is low [32]. This may impact reliability of results due to potential misclassification bias. In this cohort, 21.5% of women reported hypertension during pregnancy. This can be considered moderately high given the rates of preeclampsia (1.5–7.7%) and gestational hypertension (4.2–7.9%) in the general population. However, it is recognised that the prevalence of HDP from population health data is under-reported [33]. It is probable that not all cases of new-onset hypertension in pregnancy fit specific diagnostic criteria for HDP subtypes. For example, in the Northern Finland Birth Cohort 1966 study [34], >96.3% of births in Northern Finland were recorded and blood pressures of pregnant women measured. Among ~39% hypertensive pregnancies (~2% preeclampsia, ~9% gestational hypertension), ~15% did not fit diagnostic criteria for known HDP subtypes. Additionally, due to the limitations of the data, we were unable to discriminate between the individual associations of each of the four subtypes of HDP and the odds of cancer.

We cannot account for residual confounding and unknown confounders. Confounders were gathered at baseline, which could vary throughout the time of follow-up. However, this is unlikely to greatly influence the results, as participants were aged 39–79 years at time of study entry and many variables adjusted for are unlikely to change given the age range (all reproductive variables and socioeconomic factors such as educational status). It is possible, however, that participants had accrued additional comorbidities. To prevent participants being omitted from analyses which would lower the statistical power of our results, we included missing values as separate categories. Healthy volunteer bias may be a factor in this cohort, as this was a volunteer study requiring long-term follow-up. However, baseline characteristics of the EPIC-Norfolk have been shown to be similar to other UK representative population samples [35]. Participants were mainly White British (>99.6%); therefore, our findings may not be generalisable to populations with different ethnic backgrounds.

## 4. Materials and Methods

### 4.1. Population

The EPIC-Norfolk cohort is sampled from men and women aged 39–79 years at the time of study entry (1993–1997), in Norfolk, UK. Participants were invited through general practice registers. A total of 16,740 women gave consent and were sent health and lifestyle questionnaires (HLQs). Participants were invited to attend baseline health checks, which 14,030 women attended.

### 4.2. Measurement Methods

Details of data collection and measurement methods have been described in full previously [35]. In brief, participants completed health and lifestyle questionnaires at baseline which have common formats across the EPIC cohorts. These evaluated past medical history, lifestyle factors, physical activity, occupational social class, educational status, smoking status, alcohol consumption, use of medications and reproductive history. Biological and physiological parameters including non-fasting venous blood samples, measured height, weight and blood pressure were collected by one of seven trained research nurses adhering to EPIC protocols at baseline health check. Information on exposure of interest was obtained in the women’s section of the health and lifestyle questionnaire, where participants reported whether or not they ever had hypertension during pregnancy. Other reported information on obstetric history included gestational diabetes, age at first live birth, parity, number of stillbirths and number of miscarriages or abortions. Ethical approval was provided by the Norwich Ethics Committee (98CNO1) and all participants gave informed signed consent for participation in the study with examination of medical records and use of data.

### 4.3. Case Ascertainment

Admission and diagnostic episodes for cancer were identified from NHS hospital information system and ENCORE (East Norfolk COmmission Record). Incident cancer was identified from death certificates or hospital discharge codes using ICD-9 codes: 140–208 and ICD-10 codes: C00–C97. The ICD-9 and ICD-10 codes for independent associations between HDP and common female cancers were 174.0 and C50 for breast cancer, 153.0–153.9, 154.0–154.1, 159.0 and C18–C20 for colorectal cancer, 162.0–162.9 and C33–C34 for lung cancer, 183.0 and C56–C57 for ovarian cancer and 182.0 and C54.1 for endometrial cancer. Participants were followed up from date of study enrolment until March 2016 for incident cancer events.

### 4.4. Statistical Analysis

Baseline characteristics for women with and without incident cancer were analysed using Pearson’s chi-squared test for categorical measures and Mann–Whitney *U* tests to compare medians for non-parametric continuous variables.

Univariable logistic regression analyses were performed on all variables included in women’s baseline characteristics. A multivariable logistic regression model was used to determine the association between HDP and subsequent cancer. The model adjusted for potential confounders gathered at baseline. These included: age, education level, occupational social class, smoking status, alcohol consumption, physical activity, body mass index (BMI), family history of cancer, prevalent myocardial infarction (MI), prevalent stroke, prevalent diabetes, diabetes during pregnancy, age at first live birth, number of children, number of stillbirths and number of miscarriages or abortions. BMI was categorised according to The World Health Organization BMI categories for Europe [36]. Analyses were repeated for five common cancers in site-specific cancer analyses (breast, colorectal, lung, ovarian and endometrial). Prevalent stroke was not adjusted for in the multivariable analysis for ovarian cancer due to there being no prevalent strokes in cases, and the number of children was not adjusted for in the multivariable analysis for endometrial cancer, as none of the cases with endometrial cancer had no children. We additionally conducted separate univariable and multivariable analyses adjusting for confounders as detailed above, to determine the association of HDP and inactive physical activity on incident cancer or site-specific cancers in the subgroup of high BMI (pre-obesity to obesity class II and above).

Some variables (education level, occupational social class, smoking status, alcohol consumption, BMI, age at first live birth and number of children) had missing data. Most missing data was approximately <1%, although the variables alcohol consumption and BMI had ~15–20% missing. Missing values were dealt with using the missing-indicator method, where missing data were re-coded into separate categories within variables on SPSS and were included in statistical analyses [37]. Data were analysed using SPSS version 25.0 (SPSS Inc., Chicago, IL, USA).

## 5. Conclusions

This study found no association between HDP and future cancer risk and amongst five common cancers in women in a UK-based general practice-derived population. Current research underlining HDP and cancer risk is limited to specific populations only. Future work in different populations focusing on complex genetic and hormonal factors involved in these conditions is required.

## Figures and Tables

**Table 1 cancers-12-03100-t001:** Baseline characteristics of 13,562 women of the EPIC-Norfolk according to whether participants did or did not have incident cancer.

Baseline Characteristics	Incident Cancer		*p*-Value
	No (*n* = 10,947)	Yes (*n* = 2615)	
Median age years (IQR) (*n* = 13,562)	57.3 (16.2)	60.8 (14.8)	<0.001
Education (%) (*n* = 13,562)			<0.001
No qualification	4825 (44.1)	1272 (48.7)	
0 Level	1313 (12.0)	277 (10.6)	
A Level	3751 (34.3)	845 (32.3)	
Degree	1058 (9.7)	221 (8.5)	
Social class (%) (*n* = 13,190)			0.188
Professional	674 (6.2)	146 (5.6)	
Manager	3586 (32.8)	799 (30.6)	
Skilled non-manual	2025 (18.5)	500 (19.1)	
Skilled manual	2373 (21.7)	581 (22.2)	
Semi-skilled	1524 (13.9)	381 (14.6)	
Non-skilled	476 (4.3)	125 (4.8)	
Missing	289 (2.6)	83 (3.2)	
Smoking history (%) (*n* = 13,441)			<0.001
Never	6131 (56.0)	1347 (51.5)	
Ex-smoker	3469 (31.7)	855 (32.7)	
Currently smoking	1245 (11.4)	394 (15.1)	
Missing	102 (0.9)	19 (0.7)	
Alcohol (g) (%) (*n* = 11,035)			0.005
0	2167 (19.8)	544 (20.8)	
0.1–4.9	3535 (32.3)	787 (30.1)	
5–14.9	2524 (23.1)	550 (21.0)	
15–29.9	534 (4.9)	147 (5.6)	
≥30	195 (1.8)	52 (2.0)	
Missing	1992 (18.2)	535 (20.5)	
Physical activity (%) (*n* = 13,562)			<0.001
Active	4206 (38.4)	849 (32.5)	
Inactive	6741 (61.6)	1766 (67.5)	
BMI (kg/m^2^) ^1^ (%) (*n* = 11,414)			<0.001
Normal	4127 (37.7)	822 (31.4)	
Underweight	63 (0.6)	9 (0.3)	
Pre-obesity	3520 (32.2)	935 (35.8)	
Obesity class I	1194 (10.9)	275 (10.5)	
Obesity class II +	349 (3.2)	120 (4.6)	
Missing	1694 (15.5)	454 (17.4)	
Family history of cancer (%) (*n* = 5295)	4179 (38.2)	1116 (42.7)	<0.001
Missing	1 (0.0%)	0 (0.0)	
Prevalent myocardial infarction (MI) (%) (*n* = 200)	159(1.5)	41 (1.6)	0.455
Missing	3 (0.0)	2 (0.1)	
Prevalent stroke (%) (*n* = 144)	124 (1.1)	20 (0.8)	0.79
Missing	3 (0.0)	0 (0.0)	
Prevalent diabetes (%) (*n* = 226)	193 (1.8)	33 (1.3)	0.109
Missing	5 (0.0)	0 (0.0)	
Hypertension during pregnancy (%) (n = 2919)	2341(21.4)	578 (22.1)	0.494
Missing	1244 (11.4)	310 (11.9)	
Diabetes during pregnancy (%) (n = 219)	173 (1.6)	46 (1.8)	0.229
Missing	620 (5.7)	169 (6.5)	
Age at first live birth, years (%) (*n* = 13,304)			0.008
≤20	1779 (16.3)	395 (15.1)	
21–25	5189 (47.4)	1169 (44.7)	
26–30	2773 (25.3)	722 (27.6)	
≥31	1002 (9.2)	275 (10.5)	
Missing	204 (1.9)	54 (2.1)	
Number of children (%) (*n* = 13,562)			0.299
0	161 (1.5)	42 (1.6)	
1	1843(16.8)	479 (18.3)	
2	5149 (47.0)	1210 (46.3)	
≥3	3794 (34.7)	884 (33.8)	
Number of stillbirths (%) (*n* = 13,562)			0.918
None	10608 (96.9)	2533 (96.9)	
Any	339 (3.1)	82 (3.1)	
Number of miscarriages or abortions (*n* = 13,562)			0.802
0	8178 (74.7)	1978 (75.6)	
1	2077 (19.0)	479 (18.3)	
2	493 (4.5)	1113 (4.3)	
≥3	119 (1.8)	45 (1.7)	

^1^ BMI (kg/m^2^) categories defined as: <18.5 for underweight, 18.5–24.9 for normal weight, 25–29.9 for pre-obesity, 30–34.9 for obesity class I and >35 for obesity class II and above.

**Table 2 cancers-12-03100-t002:** Unadjusted and adjusted odds ratios and corresponding 95% confidence intervals for incident cancer.

Variables	Unadjusted OR (95% CI)	*p*-Value	Adjusted OR ^1^ (95% CI)	*p*-Value
Mean age years (*n* = 13,562)	1.03 (1.01–1.03)	<0.001	1.03 (1.02–1.03)	<0.001
Education (*n* = 13,562)		<0.001		0.796
No qualification	Ref		Ref	
0 Level	0.80 (0.69–0.92)		0.93 (0.80–1.08)	
A Level	0.86 (0.78–0.94)		0.98 (0.88–1.09)	
Degree	0.79 (0.68–0.93)		0.96 (0.80–1.14)	
Social class (*n* = 13,190)		0.189		0.865
Professional	Ref		Ref	
Manager	1.03 (0.85–1.25)		(0.82–1.22)	
Skilled non-manual	1.14 (0.93–1.40)		1.02 (0.82–1.26)	
Skilled manual	1.13 (0.93–1.38		1.10 (0.89–1.35)	
Semi-skilled	1.15 (0.93–1.43)		1.06 (0.85–1.33)	
Non-skilled	1.21 (0.93–1.58)		1.07 (0.81–1.42)	
Smoking history (*n* = 13,441)		<0.001		<0.001
Never	Ref		Ref	
Ex-smoker	1.12 (1.02–1.23)		1.08 (0.98–1.19)	
Currently smoking	1.44 (1.27–1.64)		1.57 (1.38–1.79)	
Alcohol (g) (*n* = 11,035)		0.005		0.235
0	Ref		Ref	
0.1–4.9	0.89 (0.79–1.00)		0.95 (0.84–1.07)	
5–14.9	0.87 (0.76–0.99)		0.96 (0.84–1.11)	
15–29.9	1.10 (0.89–1.35)		1.18 (0.95–1.46)	
≥30	1.06 (0.77–1.46)		1.20 (0.86–1.66)	
Physical activity (*n* = 13,562)		<0.001		0.022
Active	Ref		Ref	
Inactive	1.30 (1.19–1.42)		1.12 (1.02–1.23)	
BMI (kg/m^2^) (*n* = 11,414)		<0.001		<0.001
Normal	Ref		Ref	
Underweight	0.72 (0.36–1.45)		0.70 (0.34–1.42)	
Pre-obesity	1.34 (1.20–1.48)		1.26 (1.13–1.40)	
Obesity class I	1.16 (0.99–1.35)		1.10 (0.94–1.27)	
Obesity class II or more	1.73 (1.39–2.15)		1.73 (1.38–2.18)	
Family history of cancer (*n* = 5295)	1.21 (1.11–1.32)	<0.001	1.19 (1.09–1.30)	0.001
Prevalent MI (*n* = 200)	1.08 (0.77–1.53)	0.482	0.91 (0.64–1.29)	0.398
Prevalent stroke (*n* = 144)	0.67 (0.42–1.08)	0.261	0.54 (0.33–0.87)	0.041
Prevalent diabetes (*n* = 226)	0.71 (0.49–1.03)	0.201	0.58 (0.40–0.85)	0.021
Hypertension during pregnancy (*n* = 2919)	1.05 (0.95–1.17)	0.494	1.06 (0.95–1.18)	0.528
Diabetes during pregnancy (*n* = 219)	1.13 (0.81–1.56)	0.229	1.27 (0.91–1.78)	0.366
Age at first live birth, years (*n* = 13,304)		0.008		0.013
≤20	Ref		Ref	
21–25	1.02 (0.89–1.15)		1.02 (0.90–1.17)	
26–30	1.17 (1.02–1.34)		1.19 (1.03–1.38)	
≥31	1.17 (1.04–1.47)		1.26 (1.04–1.52)	
Number of children (*n* = 13,562)		0.299		0.741
0	Ref		Ref	
1	1.00 (0.70–1.42)		0.92 (0.44–1.93)	
2	0.90 (0.64–1.27)		0.97 (0.46–2.03)	
≥3	0.89 (0.63–1.26)		0.92 (0.44–1.93)	
Number of stillbirths (*n* = 13,562)		0.918		0.615
None	Ref		Ref	
Any	1.01 (0.79–1.29)		0.94 (0.73–1.20)	
Number of miscarriages or abortions (*n* = 13,562)		0.802		0.999
0	Ref		Ref	
1	0.95 (0.85–1.07)		1.00 (0.89–1.12)	
2	0.95 (0.77–1.17)		0.99 (0.80–1.23)	
≥3	0.94 (0.67–1.30)		0.98 (0.70–1.37)	

^1^ Adjusted for age, education level, occupational social class, smoking status, alcohol consumption, physical activity, BMI, family history of cancer, prevalent MI, prevalent stroke, prevalent diabetes, diabetes during pregnancy, age at first live birth, number of children, number of still births and number of miscarriages or abortion.

**Table 3 cancers-12-03100-t003:** Hypertension during pregnancy and unadjusted and adjusted odds ratios with corresponding 95% CIs of incident site-specific breast, colorectal, lung, ovarian and endometrial cancers.

Site-Specific Cancers	Hypertension during Pregnancy			
	Unadjusted OR (95% CI)	*p*-Value	Adjusted OR ^1^ (95% CI)	*p*-Value
Breast (*n* = 702)	1.09 (0.91–1.31)	0.237	1.06 (0.88–1.28)	0.713
Colorectal (*n* = 466)	1.18 (0.93–1.45)	0.399	1.15 (0.92–1.54)	0. 154
Lung (*n* = 247)	0.84 (0.60–1.18)	0.025	0.96 (0.68–1.35)	0.122
Ovarian (*n* = 188)	1.36 (0.98–1.90)	0.172	1.30 (0.93–1.83) ^2^	0.183
Endometrial (*n* = 163)	1.42 (0.99–2.03)	0.120	1.16 (0.80–1.67) ^3^	0.577

^1^ Adjusted for age, education level, occupational social class, smoking status, alcohol consumption, physical activity, BMI, family history of cancer, prevalent MI, prevalent stroke, prevalent diabetes, diabetes during pregnancy, age at first live birth, number of children, number of still births and number of miscarriages or abortion. ^2^ Adjusted for age, education level, occupational social class, smoking status, alcohol consumption, physical activity, BMI, family history of cancer, prevalent MI, prevalent diabetes, diabetes during pregnancy, age at first live birth, number of children, number of still births and number of miscarriages or abortion. ^3^ Adjusted for age, education level, occupational social class, smoking status, alcohol consumption, physical activity, BMI, family history of cancer, prevalent MI, prevalent stroke, prevalent diabetes, diabetes during pregnancy, age at first live birth, number of still births and number of miscarriages or abortion.

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
