# Peer review of "Hypertensive Disorders of Pregnancy (HDP) and the Risk of Common Cancers in Women: Evidence from the European Prospective Investigation into Cancer (EPIC)-Norfolk Prospective Population-Based Study"

_cancers, 2020, doi:10.3390/cancers12113100_

Round 1
Reviewer 1 Report
Pasdar et al. carried out a cohort to determine the association between hypertensive disorder during pregnancy and common cancers, in a UK population-based longitudinal study. They concluded that hypertensive disorder during pregnancy was not associated with overall maternal cancer risk. Their data include important message regarding cancer risk stratification for patients with HDP. Overall, the manuscript was well written.
Major Revision
1) Table 2 showed unadjusted and adjusted odds ratios of risk factors for incident cancer with corresponding 95% confidence intervals (95% CIs). This reviewer wonders a multicollinearity problem in the multivariable logistic regression model. Why do not select items (less than 10) in the multivariable logistic regression analyses according to the results of univariable logistic regression analyses.
2) This reviewer anxious to know difference in clinical background between participants with or without HDP. Which factors significantly associated with HDP?
3) Table 2 showed that smoking history, high BMI, or inactive physical activity is significantly associated with incident of cancer. P 11 lines 195-197: The authors described that most women with HDP belonged to the pre-obesity category (1003 women, 34.4%). This reviewer suggests to analyze association between HDP and cancer incident in the subgroup of high BMI. In addition, I suggest to analyze association between inactive physical activity and cancer incident in the subgroup of high BMI.
Minor revision
1) Page 4. What is the definition of active physical activity?
2) Page 4. The authors included silent lacunar stroke as prevalent stroke?
Reviewer 2 Report
This is large case-control registry study of over 13,000, divided into those no cancer and those with cancer. Then they looked at hypertension during pregnancy and found no associations with breast, colorectal, lung, ovarian and endometrial cancer. The limitations are well-acknowledged, mainly hypertension during pregnancy was self-reported. I find no particular fatal flaw in the manuscript but other studies including meta-analysis have reported this before and I think its' overall interest to readers is going to be low. Therefore, I advise rejection of this manuscript.
Author Response
Thank you for your comment. Other reviewers have endorsed the paper and we have considered suggestions of the reviewers. It is now much improved by the comments and suggestions. We believe this study can add to the wider literature and we hope the editors will consider accepting it.

Reviewer 3 Report
In this report Pasdar et al. describe the association of hypertensive disorders of pregnancy with several parameters including certain types of cancer. Overall, the manuscript adds value to the existing literature. This work has several limitations. However, limitations of this study have been properly described.
Minor comment:
There might be a type on the P value of Lungs in table 3.
Author Response
Thank you for your comment. This has been double-checked, and there is no typo present. The overall p-value in the univariable analysis was shown to be statistically significant due to a statistically significant missing-indicator category.

Round 2
Reviewer 2 Report
No comments